# Working Memory-Related Neurofunctional Correlates Associated with the Frontal Lobe in Children with Familial vs. Non-Familial Attention Deficit/Hyperactivity Disorder

**DOI:** 10.3390/brainsci13101469

**Published:** 2023-10-18

**Authors:** Xiaobo Li, Chirag Motwani, Meng Cao, Elizabeth Martin, Jeffrey M. Halperin

**Affiliations:** 1Department of Biomedical Engineering, New Jersey Institute of Technology, Newark, NJ 07102, USA; cm597@njit.edu (C.M.); meng.cao@njit.edu (M.C.); elizabeth.martin@njit.edu (E.M.); 2Department of Electrical and Computer Engineering, New Jersey Institute of Technology, Newark, NJ 07102, USA; 3Graduate School of Biomedical Sciences, Rutgers University, Newark, NJ 07102, USA; 4Department of Psychiatry, Icahn School of Medicine at Mount Sinai, New York, NY 10029, USA; 5Department of Psychology, Queens College, City University of New York, New York, NY 11367, USA; jeffrey.halperin@qc.cuny.edu

**Keywords:** ADHD, working memory, familial, fMRI, n-back, children, ABCD, correlation, laterality, prefrontal cortex

## Abstract

Attention deficit/hyperactivity disorder (ADHD) is a neurodevelopmental disorder with high prevalence, heritability, and heterogeneity. Children with a positive family history of ADHD have a heightened risk of ADHD emergence, persistence, and executive function deficits, with the neural mechanisms having been under investigated. The objective of this study was to investigate working memory-related functional brain activation patterns in children with ADHD (with vs. without positive family histories (ADHD-F vs. ADHD-NF)) and matched typically developing children (TDC). Voxel-based and region of interest analyses were conducted on two-back task-based fMRI data of 362 subjects, including 186, 96, and 80 children in groups of TDC, ADHD-NF, and ADHD-F, respectively. Relative to TDC, both ADHD groups had significantly reduced activation in the left inferior frontal gyrus (IFG). And the ADHD-F group demonstrated a significant positive association of left IFG activation with task reaction time, a negative association of the right IFG with ADHD symptomatology, and a negative association of the IFG activation laterality index with the inattention symptom score. These results suggest that working memory-related functional alterations in bilateral IFGs may play distinct roles in ADHD-F, with the functional underdevelopment of the left IFG significantly informing the onset of ADHD symptoms. Our findings have the potential to assist in tailored diagnoses and targeted interventions in children with ADHD-F.

## 1. Introduction

Attention deficit/hyperactivity disorder (ADHD) is a neurodevelopmental disorder with a high prevalence rate, ranging between 5.29% and 9.4% for children worldwide [1,2,3]. Clinically, ADHD may present as primarily inattentive, primarily hyperactive/impulsive, or combined type when diagnosed based on at least a 6-month presence of symptoms in multiple settings such as school and the home [4,5]. ADHD children show a notable heterogeneity in clinical and cognitive/behavioral profiles, comorbidities, developmental stages and trajectories, and neural correlates (underlying neural mechanisms associated with the given cognitive function), likely due to the heterogeneity in etiological factors (i.e., biological and environmental risk factors) [1,6,7,8]. Considerable evidence on multiple units of analysis (e.g., behavior, paradigms, physiology, circuits) have separately implicated each of the executive function (EF) and attention domains in the pathophysiology of ADHD, while the moderate effect size of these findings and the substantial proportion of children with ADHD in these studies who did not exhibit impairments prove that individual deficits in these cognitive domains cannot account for all cases of the disorder [9,10,11]. Familial risk is an important etiological factor of ADHD. Heritability estimates are as high as 60–80%, and studies have showed a five-fold risk of developing ADHD and a four-fold risk of symptom persistence into adolescence and/or adulthood in children with a positive family history of the disorder [12,13,14,15,16]. Understanding the distinct neural mechanisms underlying familial, as compared to non-familial, ADHD is urgent and critical for the development of more tailored diagnoses and neurobiologically targeted treatments/interventions in affected children.

Several studies have associated ADHD with impaired cognitive development and deficits in EF, especially working memory [17,18]. EF is a group of top-down mental processes that are responsible for focus and attention to the task at hand in aid of planning, inhibitory control, and working memory [19]. Working memory is the interim storage and manipulation of necessary information for the execution of tasks at hand [20]. Research using functional magnetic resonance imaging (fMRI) shows that functional activation during a working memory task can predict ADHD symptomatology [21]. Although EF deficits are implicated in ADHD, evidence for these deficits in ADHD is inconsistent [22]. ADHD participant groups show a notable heterogeneity in EF and working memory deficits [9,23,24,25,26]. Furthermore, inconsistencies have been noted regarding the specific EF and working memory domains in which participants with ADHD show impairments [27,28,29]. There is also inconsistency across reports of brain regions, in which fMRI activation during working memory differ between ADHD participants and typically developing children (TDC), with some studies showing the significance of occipito-parietal lobe regions, while others show frontal and temporal lobe region differences [30,31,32]. Such heterogeneity in results regarding the clinical/behavioral profiles and functional activations in ADHD may be attributed to the aforementioned underlying etiological heterogeneity in biological and environmental risk factors, and differences in methodologies across various studies [7,8,9,10,11].

Familial risk has been suggested to be an important contributor to impaired EF in ADHD children [33,34,35]. This suggestion is supported by twin and sibling studies, in which reduced working memory performance (measured using verbal and visuo-spatial tasks) was observed in ADHD children/adolescents and their unaffected twin/siblings compared to TDC [14,36,37,38]. These studies suggest an association between familial risk for ADHD and working memory deficits [39]. Thus, investigating the neural substrates of working memory deficits associated with a family risk of ADHD may shed light on differential neural mechanisms underlying familial vs. non-familial ADHD (ADHD-F vs. ADHD-NF), which can further provide guidance on targeted pharmacological and non-pharmacological treatments for children with ADHD [40,41].

Previous multimodal neuroimaging studies in siblings and parent–child pairs have implicated alterations in prefrontal cortical (PFC) regions associated with familial ADHD. Among these, a voxel-based structural MRI (sMRI) study reported medial and orbito-frontal cortex grey matter volume reductions in ADHD children and their unaffected siblings compared to TDC [42]. A multimodal (fMRI and diffusion tensor imaging) study in parent–child pairs (with and without ADHD) reported that prefrontal fiber tract measures were significantly associated between ADHD parents and their children, and fractional anisotropy in the right prefrontal fiber tracts correlated with both functional activity in the inferior frontal gyrus (IFG) and caudate nucleus, and the performance of a go/no-go task in parent–child dyads with ADHD [43]. A Stroop task-based fMRI study found that, relative to TDC, both groups of twin siblings (one with ADHD and the other unaffected) showed significantly reduced activations in the regions of the fronto-parietal network, especially the superior frontal gyrus (SFG) [44]. These studies suggest a familial influence on the structural and functional patterns of the PFC in ADHD. Nevertheless, these existing studies lacked a matched independent ADHD-NF group for comparisons.

The current study aimed to investigate working memory-related functional brain mechanisms associated with familial vs. nonfamilial ADHD in three matched and independent groups of ADHD-F, ADHD-NF, and TDC from the baseline pool of the Adolescent Brain Cognitive Development (ABCD) study. The ABCD study recruitment replicated the demographic characteristics of the general population, with similar rates of familial vs. non-familial ADHD as shown in other large-scale community-based studies [13,14,45,46]. The baseline data of the ABCD study includes 9–10-year-old participants, with 47.8% females and a racial distribution including 52.1% White, 20.3% Hispanic, 15.0% Black, 2.1% Asian, and others [47]. We hypothesized that, relative to the groups of TDC and ADHD-NF, children with ADHD-F would show distinct activation patterns associated with the PFC during working memory processes, and such neural alterations may be linked to cognitive impairment in working memory and/or ADHD symptomatology, specific to ADHD-F. Therefore, the success of this proposed study may contribute to the development of distinct neural markers of ADHD-F vs. ADHD-NF that can inform more tailored diagnoses and treatments for affected children.

## 2. Materials and Methods

### 2.1. Participants

The participants in the current study were enrolled from the baseline pool of the ABCD study, taking advantage of the large (over 11,000 participants) neuroimaging dataset of this prospective study on brain development and behavior in children and adolescents, recruited from 21 sites across the US [48,49]. This large sample size provides increased statistical power to compare groups and draw conclusions. The general exclusion criteria for the entire study sample were (1) a history or current diagnosis of bipolar disorder, schizophrenia, autism, pervasive developmental disorder and/or chronic tic disorder, or other neurological disorders; (2) a history or current diagnosis of traumatic brain injury; (3) a family history of psychiatric disorders in biological parents, including autism, bipolar disorder, schizophrenia, psychosis (of any type), or current diagnoses based on the Mini International Neuropsychiatric Structured Interview; (4) a current prescription of psychostimulant medication or a history of substance abuse; and (5) an intelligence quotient (IQ) of less than 80 [50].

ADHD symptomatology in the ABCD study was assessed using the recently validated and computerized Kiddie Schedule for Affective Disorders and Schizophrenia (KSADS-5), an interview designed to assess symptomatology across a wide range of psychiatric disorders [51,52,53]. KSADS-5 has been rigorously studied for validation and reliability in multiple research and clinical settings among children and adolescents [54,55]. Each symptom is evaluated on a 5-point scale: 0 (not present), 1 (rarely), 2 (several days), 3 (more than half the days), and 4 (seen every day). A response of 2 or more was considered as a positive endorsement for that symptom. ADHD presentation (i.e., inattentive, hyperactive–impulsive, combined) was determined using the Diagnostic and Statistical Manual (version 5) criteria. Subjects who were endorsed for 6 or more symptoms out of 9 for the inattention symptom count were considered as having inattentive presentation; 6 or more symptoms out of 9 for the hyperactivity symptom count was considered as a hyperactive/impulsive presentation; and 6 or more symptoms out of 9 for both domains was considered as a combined presentation of ADHD. Subjects with any of the three current presentations of ADHD were included in the ADHD group.

Among those with ADHD, the biological parent history of ADHD was assessed based on the parents’ medical history questionnaire and the Conners Adult ADHD Rating Scale (CAARS) that was administered to parents during the baseline study visit [56]. A child with ADHD-F was defined as having at least one biological parent with a reported diagnosis of ADHD and/or a T-score of >65 on at least one of the three CAARS subscales for inattentive symptoms, hyperactive/impulsive symptoms, or ADHD symptoms in total. A child with ADHD-NF was defined as having no reports of ADHD diagnoses from both biological parents and T-scores of <60 on all the three subscales of the CAARS.

Among 588 children with ADHD who met the general and group-specific inclusion and exclusion criteria described above, 412 were excluded due to insufficient parental history information; missing sMRI data; neuroimaging artifacts; heavy head motion in fMRI scans; an n-back task (both 0-back and 2-back) performance accuracy of less than 65%; missing fMRI scans, or missing task performance data. Thus, a total of 176 children with ADHD (80 ADHD-F and 96 ADHD-NF) participated in further analyses.

A total of 186 children without a diagnosis of ADHD comprised the TDC group. The groups of TDC and ADHD were well matched for age, sex, race, handedness (using the short version of the Edinburgh handedness inventory [57]), IQ, pubertal developmental scale (using the ABCD Youth Pubertal Development Scale and Menstrual Cycle Survey History [58]), as well as education and the combined income of parents.

Briefly, the participants of this study were obtained from the ABCD study baseline pool. The general inclusion/exclusion criteria were first applied to all the baseline subjects with or without ADHD. The imaging and task performance data quality were examined to further exclude unqualified subjects. Then, among candidates in the group of ADHD, the medical histories of their biological parents were assessed to form the sub-groups of ADHD-F and ADHD-NF and remove subjects with unclear family risk information. Participants in the TDC group were chosen to best match with the group of ADHD for age, sex, race, handedness, IQ, pubertal developmental scale, parental education, and combined income. Further information on the participant characteristics can be found in Table 1.

The CBCL attention problem T-scores are the normalized scores from the attention CBCL syndrome scale, and CBCL ADHD T-scores are the normalized scores from the ADHD CBCL DSM-5 scale [59]. Both the KSADS-5 measures were obtained from the questionnaire, with 9 questions for each measure. Each question is a representative of a symptom of the respective domain (inattention or hyperactivity/impulsivity) of ADHD.

### 2.2. The n-Back Task

A modified n-back task with a block-based design was used in the ABCD study’s fMRI protocol, which has been detailed previously [60]. The n-back is robust in eliciting activation in brain regions associated with attention and working memory [61]. The task consisted of two conditions, 0-back and 2-back. In the 0-back block, the subjects were shown a target image at the start of the block and asked if the trial stimulus matched the target. In the 2-back block, the subjects were instructed to respond if an image matched the image that was shown 2 trials before. The subjects responded using two buttons, one button indicating “match” and the other button indicating “no match”.

This block-based design spanned over a set of 2 runs of 8 blocks each, with an equal number of blocks for each condition (0-back/2-back, 4 blocks per condition per run), as shown in Figure 1. The order of blocks was randomized into 4 sets and these sets were counterbalanced for each subject. The fixation cross (for 0.5 s) and instruction cue (for 2.5 s) were placed at the start of each block, followed by a sequence of 10 trials (2.5 s per trial). Each trial consisted of a visual stimulus (for 2 s) followed by a fixation cross (for 0.5 s). The visual stimuli included faces (happy, neutral, fearful) and places. Each block lasted for 28 s. A fixation cross was placed after every 2 task blocks and at the end of each run (for 15 s and 5 s, respectively). Each run lasted for 289 s.

### 2.3. Image Acquisition

The protocols and parameters of the ABCD study neuroimaging data acquisition have been previously described in detail [60]. Briefly, fMRI data from each of the 362 subjects were acquired on either a Siemens, General Electric (GE), or Philips scanner, with a field strength of 3T using simultaneous multi-slice/echo-planar imaging sequences with the following parameters: a resolution of 2.4 mm isotropic, a repetition time (TR) of 800 ms, an echo time (TE) of 30 ms, a 90 × 90 matrix, and 60 slices per volume with a 216 × 216 field of view (FOV) [60,62]. The number of volumes varied with the scanner machine [62]. The raw data from the Siemens and Philips scanner machines had 370 volumes each for each run, whereas the raw data from the GE-DV25 and the GE-DV26 had 367 and 378 volumes, respectively.

Three-dimensional T1-weighted sMRI data of each of the 362 subjects were acquired on a 3T scanner (either Siemens, GE, or Philips) with a 32-channel head coil. The sMRI data were acquired using inversion-prepared RF-spoiled gradient echo pulse sequences, with the following parameters: a voxel size of 1.0 mm isotropic; flip angle = 8°; a TR/TE of 2500/2.88 for Siemens, 6.31/2.9 for Philips, and 2500/2 for GE scanners; FOV = 256 × 256 for the Siemens and GE scanners and 256 × 240 for the Philips scanner.

### 2.4. Individual-Level fMRI Data Preprocessing

For each subject, the fMRI data were preprocessed using the fMRI Expert Analysis Tool (FEAT) from the fMRI Software Library (FSL). Eight initial volumes were excluded for the 4D volumes collected using the Siemens Prisma Fit and Philips scanner machines. Five and sixteen initial volumes were excluded for those collected using the GE–DV25 and GE–DV26, respectively, because these initial volumes made up the pre-scan reference for the 4D images [62]. The fMRI images were slice-time corrected and motion corrected using the cost function apodization and hybrid global–local optimization method [63]. The six translational and rotational displacement parameters were utilized to measure the head motions. Subjects were excluded if they had excessive movement in a run (maximum absolute movement of >3 mm or a mean frame-wise displacement of >0.3 mm, which is an accepted and stringent cut-off for analyses based on the ABCD study cohort [60]). This was followed by brain extraction, co-registration using the sMRI of the same subject, and normalization to an asymmetric brain template of 7- to 11-year-old pediatric subjects from NeuroImaging and the Surgical Technologies Lab [64,65]. Further, the fMRI images were intensity normalized, spatially smoothed with a 5 mm full-width at a half-maximum Gaussian kernel to improve the signal-to-noise ratio, and high-pass filtered with a cutoff of 85 s to remove the low frequency noise and signal drifting. Voxel-based activation maps responding to the 0-back, 2-back, and 2-0 contrast conditions were modeled using the general linear model (GLM) approach in the FEAT from the FSL [66]. The 24 motion parameters, including the 6 standard motion parameters, their time derivatives, and squares, were added as nuisance regressors. The cluster-based method for multiple comparison corrections was applied to threshold the voxel-based activation maps, with Z ≥ 2.3 at α ≤ 0.05. This cluster-based method utilized the Z-statistic threshold to define contiguous clusters, the significance level of which (from the Gaussian random field theory) was compared with the probability threshold. The Z-threshold range from 2.3 to 3.1 corresponds to the primary threshold range of *p*-values from 0.01 to 0.001, and was therefore considered to be standard in the fMRI studies [67].

### 2.5. Region of Interest (ROI) Selection and Definition

The PFC is one of the core components of the brain pathways for working memory processing, and has been frequently reported to be involved in the neurophysiology of ADHD [68]. Voxel-based fMRI studies have reported functional deficits in multiple clusters of the PFC in children with ADHD [69]. Based on the results of these existing studies and the voxel-based results of the current study, three bilateral ROI pairs were identified in regions of the PFC. These ROIs showed significant between-group differences (either ADHD vs. TDC or ADHD-F vs. ADHD-NF) of the voxel-based 2-0 contrast activations. These voxel-based intermediate group comparisons were conducted using the GLM and were controlled for sex, handedness, IQ, race, pubertal status, scanner machine, parental income, and education status using the FEAT from the FSL [70]. Cluster corrections were applied for multiple comparisons with thresholds of Z ≥ 2.3 and α ≤ 0.05.

The ROI masks were then created by overlapping the activation maps of the TDC group with the automated anatomical labeling atlas version 3 [71,72]. Then, for each subject, the 2-0 contrast activation magnitude of all the voxels in each ROI were averaged to generate the ROI-specific activation values.

### 2.6. Laterality Analyses

Lateralized activation patterns for cognitive information processes have been frequently observed in fMRI studies in human subjects [73,74,75,76]. The laterality index (LI) has been applied as a metric for describing the hemispherical domination patterns of fMRI activation [77,78]. In this study, the LI of activation in each bilateral ROI pair was evaluated using Equation (1):(1)$$LI=\frac{L-R}{\left|L\right|+\left|R\right|}$$
where *L* and *R* are the ROI-specific activation t-values in the left and right hemispheres, respectively. The LI is used to study hemispherical patterns of brain activation for a given brain region, where a positive LI value indicates a left-hemisphere-dominated activation pattern, and negative LI value indicates a right-hemisphere-dominated activation pattern.

### 2.7. Statistical Analyses

Descriptive statistics were performed in SPSS (IBM Corp. Released 2020. Version 27.0. Armonk, NY, USA) using an independent samples t-test (for continuous variables) and chi-square analysis (for discrete variables) to study the group differences between TDC vs. ADHD and between ADHD-NF vs. ADHD-F for demographic measures, the pubertal developmental scale, behavioral measures, and task performance (0-back accuracy, 2-back accuracy, 0-back mean reaction time (RT), and 2-back mean RT). Levene’s test was used to check for equality of variance.

ROI-based activations and the LI were analyzed for group differences among TDC, ADHD-F, and ADHD-NF using a one-way analysis of variance (ANOVA). Aiming to balance the sample sizes of the groups of TDC, ADHD-F, and ADHD-NF, as well as cross-validate the group comparison results, subjects in the TDC group were randomly split into two sub-samples for testing and validation (TDC-1, N = 91 and TDC-2, N = 95). A post hoc analysis was conducted using an independent samples t-test between each group pair.

Understanding the brain–behavioral relationship is critical for searching the neural mechanisms that are associated with cognitive impairments and clinical symptoms in brain disorders [79]. In our brain–behavioral correlation analyses for ADHD-F and ADHD-NF, the mean activation of each ROI and the LI were analyzed for correlation with ADHD symptomatology and task performance measures using two-tailed Pearson correlation in SPSS version 27. The Child Behavior Checklist (CBCL) ADHD T-scores, the CBCL attention T-scores, the KSADS-5 inattention raw scores, and the KSADS-5 hyperactivity raw scores were used as measures of ADHD symptomatology. The KSADS-5-based symptom raw scores were calculated based on the sum of answers to the 9 questions in each subscale of inattention or hyperactivity/impulsivity. The task performance measures included the percentage accuracy and mean RT for each task condition (0-back and 2-back). Bonferroni correction, α = 0.05/2 = 0.025, was applied for each ROI pair. The steps of the methodology are summarized in Appendix A.

## 3. Results

### 3.1. Demographic and Task Performance Measures

There were no significant group differences in demographic variables between TDC and ADHD groups and between ADHD-F and ADHD-NF, as shown in Table 1. Compared to TDC, the ADHD group showed significantly increased CBCL attention problems, ADHD T-scores, and KSADS-5 inattention and hyperactivity raw scores.

Group comparisons between the TDC and ADHD groups based on task performance measures did not show significant differences (Appendix A). ADHD-F showed significantly higher 2-back accuracy (t(284) = 2.216, *p* = 0.027, d = 0.261) compared to ADHD-NF. Group comparisons between ADHD-F and ADHD-NF did not show significant differences in the 0-back and 2-back mean RT and 0-back percentage accuracy.

### 3.2. ROI-Based Activation and LI Analyses

Within the PFC, the SFG medial, middle frontal gyrus (MFG), and IFG triangular showed significant group differences (ADHD vs. TDC and/or ADHD-F vs. ADHD-NF) in the intermediate voxel-based analyses (detailed in Appendix A). ROI pairs were subsequently located in these regions (Figure 2).

A one-way ANOVA and post hoc analyses revealed that relative to TDC, children with ADHD-F showed significantly reduced mean activation during working memory processing in the left IFG (*p* < 0.001, d = 0.387) and the bilateral SFGs (left *p* = 0.002, d = 0.334; right *p* = 0.006, d = 0.29), whereas children with ADHD-NF showed significantly reduced mean activation in the bilateral IFGs (left *p* < 0.001, d = 0.367; right *p* < 0.001, d = 0.398) and the right SFG (*p* = 0.012, d = 0.258) (Figure 3 and Appendix A). Although the LIs of the three ROI pairs did not show significant differences among the three groups, the group of ADHD-F had a trend towards a right-hemisphere-shifted LI that the TDC and ADHD-NF groups did not show.

### 3.3. Brain–Behavior Correlation Analyses

In children with ADHD-F, the mean activation in the left IFG was significantly positively correlated with the 2-back RT (*p* = 0.003), the mean activation in the right IFG was significantly negatively correlated with the CBCL ADHD T-score (*p* = 0.021), and the LI in the IFG was significantly negatively correlated with the KSADS-5 inattention raw score (*p* = 0.001). In children with ADHD-NF, the LI of the IFG showed a significant positive correlation with the KSADS-5 hyperactivity raw score (*p* = 0.037); the mean activation in the left SFG showed a significant negative correlation with the CBCL attention problem T-score (*p* = 0.025) (Figure 4).

## 4. Discussion

The present study examined working memory-related PFC activation and laterality patterns in children with familial vs. non-familial ADHD, in three matched and independent groups of children with ADHD-F, ADHD-NF, and TDC. Relative to TDC, children with ADHD-F showed significantly reduced working memory-related activation in the left IFG, and this reduced left IFG activation was significantly correlated with a longer RT for 2-back task performance in ADHD-F. Relative to the groups of TDC and ADHD-NF, children with ADHD-F also had the trend of a right-hemisphere-shifted LI, which was caused by the significant dysfunction of the left IFG during working memory processing. Bounded superiorly by the inferior frontal sulcus and inferiorly by the lateral sulcus, the IFG forms an integral component of the ventrolateral PFC (VLPFC) [81]. Studies have consistently indicated that the IFG is actively involved in multiple sensory and cognitive processes including attention, working memory, and cognitive control [82,83,84]. Specifically, the IFG has been found to serve as an important component for spatial working memory processing, through its role in rapidly adapting attentional control to respond to current stimuli and retrieving stored information [85,86,87,88,89,90]. Further, studies involving EF tasks have characterized ADHD children as having long and variable RTs, with medium to large effect sizes [9,91,92]. Notably, the longer RTs in ADHD have been associated with familial risk, with familial influences as high as 72% for the relationship between RTs and ADHD phenotype [10,92].

Furthermore, we found that the mean activation of the right IFG in children with ADHD-F was significantly negatively correlated with their CBCL ADHD T-score with a small effect size, suggesting that the robust right IFG activation in this group links to reduced ADHD symptoms. Meanwhile, the LI of the IFG in the group of ADHD-F was significantly negatively correlated with the KSADS-5 inattention raw score, with a magnitude of small to medium size. The LI is a measure of the spherical weight of brain activation. The lower IFG LI in the group of ADHD-F resulted from a reduced left-side activation, relative to the right-side activation. Therefore, the negative correlation of a lower IFG LI with a higher KSADS-5 inattention score can further suggest a linkage of the left IFG hypoactivation to elevated inattentive symptoms in ADHD-F. In addition, the LI of the IFG showed an unique asymmetrical, right-higher-than-left pattern in the group of ADHD-F. Although the group differences of the LI of the IFG were not significant after Bonferroni corrections, future studies in different samples can further test if such a pattern exists in ADHD-F. Over the decades, the IFG has been implicated in attention processes by being a part of cingulo-frontal–parietal cognitive-attention network, the abnormalities in which have been widely suggested to be responsible factors in childhood ADHD [93]. Our results of reduced left IFG activation and its correlation with ADHD symptomatology are in accordance with the ADHD model of fronto-striatal pathway dysfunction, related to inattention symptoms, inhibition problems, impulsivity, and hyperactivity [94]. The dysfunction of the fronto-striatal pathway is consistent with the fact that the IFG sends fibers to the dopamine-rich putamen, the deficits in which associate with ADHD symptoms in children [95]. Meanwhile, robust activation in the right IFG (as a part of the VLPFC) has been suggested to respond to visuo-spatial stimuli, and a possible compensatory mechanism of the right IFG in children with ADHD has also been suggested in existing meta-analysis [81]. Therefore, together with the existing IFG-related findings in ADHD, our results further suggest that the functional underdevelopment of the left IFG may be associated with working memory impairment and symptomatology, whereas the right IFG may play a critical compensatory role in children with ADHD who have a positive familial history.

When the family history risk factor was not present, our results indicate that children with ADHD (i.e., the ADHD-NF group) have distinct patterns of IFG activation, laterality, and their linkages with the cognitive impairment and clinical manifestations of ADHD. Compared to TDC, children with ADHD-NF showed significantly reduced IFG activation of a medium magnitude in both hemispheres, while ADHD-F only showed reduced IFG activation in the left hemisphere, also with a medium magnitude. Existing meta-analyses of task-based fMRIs showing robust hypoactivation in the left IFG in ADHD is consistent with our results from both ADHD groups, but do not show right IFG hypoactivation, as seen in our sample of ADHD-NF [96,97]. Due to limitations on accessible information and the unknown risk factors of ADHD, it is hard to determine if the ADHD-NF group was the more or less heterogeneous set of children in terms of other causal risk factors. Therefore, it is hard to interpret if the working memory-related right IFG dysfunction in this sample of ADHD-NF is a reliable distinct substrate of all types of non-familial ADHD, or a sample-specific phenomenon.

In addition, we observed that the ADHD-NF group showed very similar non-negative LIs of IFG activation to those in TDC, and a higher IFG LI was significantly correlated with an increased KSADS-5 hyperactivity raw score in ADHD-NF with a small effect. Given that the activations in the bilateral IFGs were both significantly decreased in ADHD-NF, the maintenance of a non-negative IFG LI and its association with increased KSADS-5 hyperactivity may both be directly contributed to by the significantly reduced right IFG activation. Again, further research is needed to identify if this is a critical neural marker of all types of ADHD-NF. Putting together all the results of the IFG, the distinct patterns of working memory-related IFG activation and laterality in children with ADHD-F vs. ADHD-NF may significantly imply the unique family heritage risk factor-related neural substrate of ADHD in children. These findings therefore have the potential to serve as biomarkers for more tailored diagnoses and targeted treatments and interventions.

Compared to TDC, children with ADHD-F also showed significantly reduced activation in the bilateral SFGs, whereas children with ADHD-NF only showed significantly reduced activation in the right SFG. In contrast to the significant differences in the IFG, these differences had small effect size. And in children with ADHD-NF, the mean activation in the left SFG showed a significant negative correlation with the CBCL attention problem T-score, also with small magnitude. The SFG is involved in various EFs of planning, inhibition, abstract reasoning, and working memory by being a part of the dorsolateral PFC and its connections to a multitude of subcortical and cortical brain regions, such as the basal ganglia, hippocampus, thalamus, parietal, and temporal areas [98,99]. Consistent activation of the SFG during 2-back working memory tasks has been shown in meta-analysis, with SFG activation implied in the continuous updates and maintenance of the working memory during tasks that require focused attention on stimulus features [100,101]. Although SFG functional alterations are frequently associated with working memory-related deficits in ADHD, the hemispheric localization of these SFG activation deficits is inconsistent with some studies implying bilateral SFG deficits, while others suggest either left- or right-sided activation deficits in the SFG [25,102,103]. The SFG-related discrepancy of existing findings in children with ADHD may be due to sampling variations, varying diagnostic criteria, task differences, techniques of data acquisition and analyses, and other factors associated with different studies [9,10,11]. Adding into the literature, our study indicates that when these potential factors are controlled for, ADHD-F and ADHD-NF demonstrate distinct patterns of SFG functional abnormalities for working memory processing. It thus suggests that familial risk factors for ADHD may significantly influence the inconsistency in clinical, behavioral, and neuroimaging studies of ADHD.

There are certain limitations to the current study that need to be considered. Our study included both male and female subjects. Although it is unestablished yet whether the neurofunctional basis of familial ADHD has sex differences, differences in symptomatic and comorbidity profiles have previously been observed in clinical studies [104]. To partially omit the effects related to sex, we included sex as a fixed-factor covariate in group analyses after matching the groups for sex. Nevertheless, future research should explicitly assess the possibility that familial risk may affect boys and girls with ADHD differently. In addition, given the choice of the neuropsychological tests and measures adopted in the ABCD study, our study utilized picture vocabulary and matrix reasoning scores as metrics for IQ, instead of full-scale IQ. Furthermore, working memory processes are subserved by widespread brain regions, including distinct activation patterns in the fronto-parietal regions, along with a considerable heterogeneity in connectome organization [26,89,105]. Therefore, future research adopting a network analysis approach and assessing topological features that may differ between ADHD-F and ADHD-NF would benefit the field. The network-based approach can provide a system-level understanding of the inter-regional functional interactions during cognitive information processing and their association with ADHD-related deficits and familial risk. Another limitation arises from the use of parametric statistical models in this study, which have a restricted capacity to simultaneously investigate multiple variables and their interactions. Future research adopting machine learning strategies may effectively extract the high-dimensional information and translate the complex neuroimaging patterns into clinical significance. Finally, the cross-sectional design of this study included a single time point of 9–11-year-old children. A longitudinal study that further investigates early-onset neural biomarkers into adolescent and adulthood will help us to understand the developmental trajectory of ADHD-related functional differences in those with vs. without a familial risk of ADHD. The outcomes of such studies can significantly inform the early brain markers for different developmental trajectories in children with ADHD, and guide early treatment and intervention strategies.

On the basis of our findings, another future study direction can focus on testing treatment efficacy on left IFG functional improvement in children with ADHD-F. Previous fMRI studies have reported the normalization of IFG activation during a range of EF tasks in youth with ADHD after the administration of methylphenidate (MPH) [106]. One study, using functional near-infrared spectroscopy, reported the acute normalization of right IFG activation during a go/no-go task with MPH administration in ADHD children [107], while the absence of the effect of MPH on IFG activation has also been reported during an n-back study [30]. Over the inconsistency of these existing findings that may have resulted from the heterogeneity of ADHD, it is critical to test if MPH treatment can improve left IFG function in children with ADHD-F.

## 5. Conclusions

Our study on matched and independent groups of ADHD-F, ADHD-NF, and TDC found that children with ADHD-F have some unique working memory processing-related activation and laterality patterns in the IFG, when compared to the cohorts of TDC and ADHD-NF. Specifically, children with ADHD-F had significantly reduced working memory-related activation in the left IFG, which was significantly correlated with longer RTs for 2-back task performance. Children with ADHD-F also had a trend of right-hemisphere-shifted LIs, caused by the significant dysfunction of the left IFG during working memory processing. Meanwhile, the activation of the right IFG in children with ADHD-F was significantly negatively correlated with their CBCL ADHD T-scores, and the LI of the IFG was significantly negatively correlated with their KSADS-5 inattention raw scores. These unique patterns suggest that functional underdevelopment associated with the left IFG may serve as a distinct role in working memory impairment and symptom onset in ADHD-F. These findings have the potential to indicate critical biomarkers of ADHD-F and aid in the development of more tailored diagnostic and treatment strategies in children with ADHD.

## Figures and Tables

**Figure 1 brainsci-13-01469-f001:**
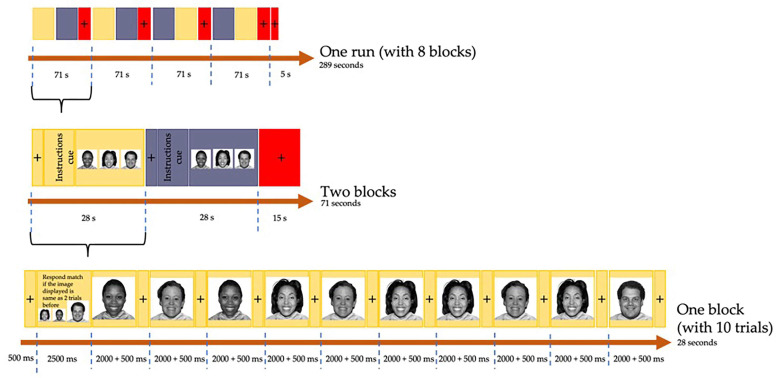
The n-back task structure (with an example of 2-back task). Grey, 0-back block; Yellow, 2-back block; +, fixation cross.

**Figure 2 brainsci-13-01469-f002:**
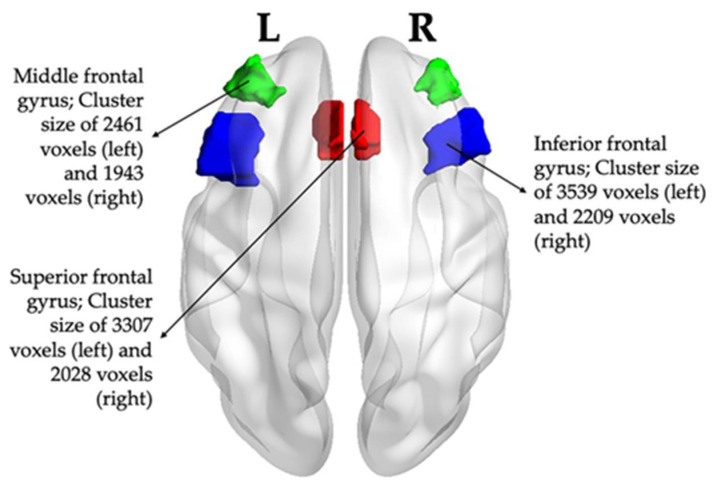
Locations of the ROI pairs. Transverse view from the top, visualizing the locations and cluster size of ROIs using BrainNet Viewer [80] (ROI: region of interest).

**Figure 3 brainsci-13-01469-f003:**
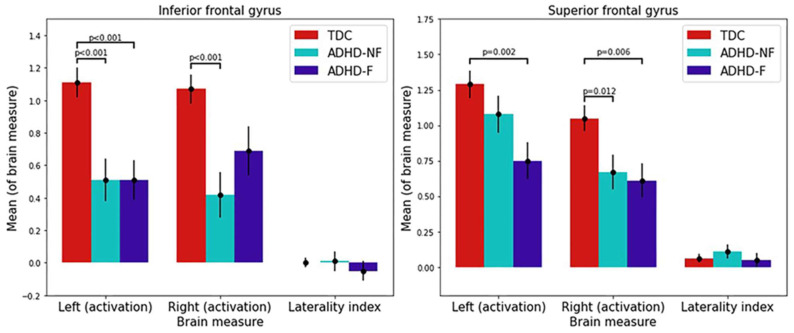
Group comparisons of ROI-based brain activation and laterality. Each bar shows the mean value involving all the subjects of each respective group (the mean values presented here for the TDC group include subjects of both sub-samples). (ROI: region of interest; TDC: typically developing children; ADHD: attention deficit/hyperactivity disorder; ADHD-NF: non-familial ADHD; ADHD-F: familial ADHD).

**Figure 4 brainsci-13-01469-f004:**
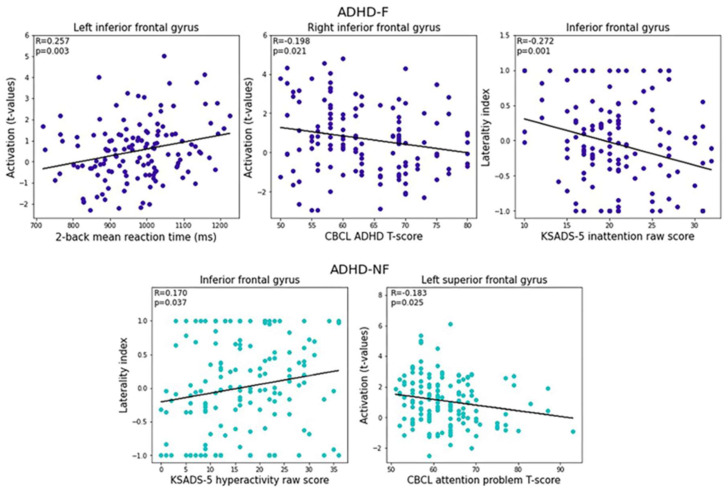
Brain–behavior correlation analyses in groups of ADHD-F and ADHD-NF (ADHD: attention deficit/hyperactivity disorder; ADHD-NF: non-familial ADHD; ADHD-F: familial ADHD; CBCL: Child Behavior Checklist; KSADS-5: Kiddie Schedule for Affective Disorders and Schizophrenia).

**Table 1 brainsci-13-01469-t001:** Group comparisons of demographic and behavioral measures (TDC: typically developing children; ADHD: attention deficit/hyperactivity disorder; CBCL: Child Behavior Checklist; KSADS-5: Kiddie Schedule for Affective Disorders and Schizophrenia; DSM-5: Diagnostic and statistical manual (version 5); IQ: intelligence quotient; N: number; SD: standard deviation).

	TDC (N = 186)	ADHD (N = 176)	Test Statistic	*p*
Sex			χ^2^(1) = 3.528	0.06
Female	89	67		
Male	97	109		
Race			χ^2^(3) = 0.898	0.826
White American	144	133		
Black American	17	16		
Mixed	17	21		
Other	8	6		
Handedness			χ^2^(2) = 1.222	0.543
Right	157	141		
Left	9	12		
Both	20	23		
Age (in months) (Mean (SD))	119.87 (6.82)	119.09 (7.84)	t(347.029) = 1.008 ^+^	0.314
Parents combined income (Mean score (SD))	7.70 (2.19) (N = 176)	7.67 (1.98) (N = 167)	t(341) = 0.150	0.881
Primary parent education status (in years) (Mean (SD))	16.35 (2.19)	16.48 (2.04)	t(360) = −0.550	0.583
IQ				
Picture vocabulary (Mean (SD))	108.18 (15.15) (N = 184)	108.96 (18.92) (N = 174)	t(331.293) = −0.426 ^+^	0.670
Matrix reasoning (Mean (SD))	10.45 (2.74) (N = 180)	9.91 (3.08) (N = 174)	t(352) = 1.749	0.081
Pubertal developmental scale (Mean score (SD))	1.61 (0.77)	1.58 (0.76)	t(360) = 0.347	0.729
Behavioral measures				
CBCL attention problems T-score (Mean (SD))	50.50 (1.051)	63.55 (8.172)	t(180.484) = −21.024 ^+^	**<0.001**
CBCL ADHD T-score (Mean (SD))	50.28 (0.798)	62.57 (7.911)	t(178.373) = −20.501 ^+^	**<0.001**
KSADS-5 inattention score (Mean (SD))	0.763 (1.466)	20.835 (5.381)	t(199.518) = −47.839 ^+^	**<0.001**
KSADS-5 hyperactivity score (Mean (SD))	0.194 (0.611)	17.761 (8.6315)	t(176.659) = −26.938 ^+^	**<0.001**

+ Levene’s test for homogeneity of variance (centered on mean) showed a significant difference in variance between the two groups. Bold *p*-values represent statistically significant group differences.

## Data Availability

Data are available upon reasonable request to the corresponding author.

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
