# Peer review of "Working Memory-Related Neurofunctional Correlates Associated with the Frontal Lobe in Children with Familial vs. Non-Familial Attention Deficit/Hyperactivity Disorder"

_brainsci, 2023, doi:10.3390/brainsci13101469_

Round 1

Reviewer 1 Report

1.ADHD-F showed asymetrical fMRI in the left and right IFG, and based on your findings, it seems that the differences can be used as an indicator for the ADHD-F and ADHD-NF. 

2. If left IFG  dysfunction is one of the possible mechanisms underlying ADHD-F, I am curious about the MPH, such as ritalin, whether the treatment of ritalin can improve the functions of impaired left IFG. 

Author Response

Manuscript ID: brainsci-2647904

Title: Working Memory-related Neurofunctional Correlates associated with Frontal Lobe in Children with Familial vs. Non-Familial Attention Deficit/Hyperactivity Disorder

We appreciate the very detailed and constructive comments from the reviewers. We have addressed all the comments and have made specific changes accordingly in the revised manuscript. Our point-by-point response to each comment is as follows in Red:

Reviewer 1

R1.1. ADHD-F showed asymmetrical fMRI in the left and right IFG, and based on your findings, it seems that the differences can be used as an indicator for the ADHD-F and ADHD-NF. 

We greatly appreciate this kind reminder. According to Supplementary Table 3 and Figure 3, the controls and ADHD-NF showed more symmetrical activations in IFG, while ADHD-F showed a significantly right-higher-than-left asymmetrical pattern of activation in IFG. We were hesitant to claim that this can be a distinct indicator of ADHD-F because the group differences of the laterality-index of IFG were not significant (after Bonferroni correction). But based on this review comment, we further strengthened our explanations of this finding in the section of Discussions, page 11: “…In addition, the LI of IFG showed an unique asymmetrical, right-higher-than-left, pattern in the group of ADHD-F. Although the group differences of the LI of IFG were not significant after Bonferroni corrections, future studies in different samples can further test if such pattern exists in ADHD-F…”

R1.2. If left IFG dysfunction is one of the possible mechanisms underlying ADHD-F, I am curious about the MPH, such as ritalin, whether the treatment of ritalin can improve the functions of impaired left IFG. 

We are grateful for this instructive comment. Although it is beyond the scope of our study, we have addressed it in Discussions, pages 12 and 13: “On the basis of our findings, another future study direction can focus on testing treatment efficacy on left IFG functional improvement in children with ADHD-F.  Previous fMRI studies have reported normalization of IFG activation during a range of EF tasks in youth with ADHD after the administration of methylphenidate (MPH) [106]. One study, using functional near-infrared spectroscopy, reported acute normalization of right IFG activation during a go/no-go task with MPH administration in ADHD children [107]. While absence of effect of MPH on IFG activation has also been reported during n-back performance [30]. Over the inconsistency of these existing findings that may have been resulted from the heterogeneity of ADHD, it is critical to test if MPH treatment can improve left IFG function in children with ADHD-F...”

  1. Kobel, M.; Bechtel, N.; Weber, P.; Specht, K.; Klarhofer, M.; Scheffler, K.; Opwis, K.; Penner, I.K. Effects of methylphenidate on working memory functioning in children with attention deficit/hyperactivity disorder. Eur J Paediatr Neurol 2009, 13, 516-523, doi:10.1016/j.ejpn.2008.10.008.
  2. Czerniak, S.M.; Sikoglu, E.M.; King, J.A.; Kennedy, D.N.; Mick, E.; Frazier, J.; Moore, C.M. Areas of the brain modulated by single-dose methylphenidate treatment in youth with ADHD during task-based fMRI: a systematic review. Harv Rev Psychiatry 2013, 21, 151-162, doi:10.1097/HRP.0b013e318293749e.
  3. Monden, Y.; Dan, H.; Nagashima, M.; Dan, I.; Tsuzuki, D.; Kyutoku, Y.; Gunji, Y.; Yamagata, T.; Watanabe, E.; Momoi, M.Y. Right prefrontal activation as a neuro-functional biomarker for monitoring acute effects of methylphenidate in ADHD children: An fNIRS study. Neuroimage Clin 2012, 1, 131-140, doi:10.1016/j.nicl.2012.10.001.

Reviewer 2 Report

I wish to express my genuine gratitude for your submission of the article. I have meticulously examined its contents and have offered constructive feedback with the intention of elevating its overall quality. I kindly request that you consider implementing the proposed revisions and adjustments. Upon your completion of these revisions, I would be pleased to conduct a further review of the revised version!

Abstract

  1. Data and Sample Size: Mentioning the sample size (362 subjects) is crucial for transparency and context. However, it would be beneficial to include information about the age range of the participants and any other relevant demographic details.
  2. Hypotheses or Research Questions: The abstract should briefly outline the hypotheses or research questions that guided the study. This helps readers understand the purpose and focus of the research.
  3. Methods Overview: Provide a brief overview of the methods used in the study, such as the specific tasks used to assess working memory and the neuroimaging techniques employed (e.g., fMRI).
  4. Results Summary: While the abstract mentions the key findings, it could be more specific. For instance, it could briefly describe the direction and significance of the observed associations in the ADHD-F group (e.g., "significant positive association" with task reaction time) to give readers a better understanding of the results.
  5. Avoid Jargon: While some level of technical language is expected in an academic abstract, it's essential to strike a balance. Avoid overly technical jargon that may alienate readers who are not experts in the field.

Introduction

  1. Research Gap: Explicitly state the research gap or the problem that your study addresses. Why is it important to investigate familial vs. non-familial ADHD? What specific questions or hypotheses will your study answer? This will help readers understand the significance of your research.
  1. Demographic Details: While you mention that the ABCD Study replicates demographic characteristics of the general population, consider providing a concise summary of these characteristics, such as age, gender distribution, and any other relevant factors, to give readers a better understanding of the study population..
  2. Transition Sentences: Use transition sentences or phrases to smoothly connect different sections of the introduction. For example, after discussing the heritability of ADHD (paragraph starting with "Heritability estimates..."), you can transition into the next paragraph by explaining how familial risk relates to this heritability.
  3. Concept Definitions: Define key terms and concepts, such as "executive function," "working memory," and "neural correlates," to ensure that readers from various backgrounds can follow your argument without ambiguity.
  4. Rationale for the Study: Provide a more explicit rationale for why investigating working memory-related mechanisms in familial vs. non-familial ADHD is essential. How might these findings contribute to our understanding of ADHD etiology and treatment?

For enrichment of your intorductionpleas euse the below refences:
https://www.brainstimjrnl.com/article/S1935-861X(23)00171-7/fulltext

https://brieflands.com/articles/ijpbs-108390.html

Materials and Methods

  1. Participant Selection: While you provide information about participant exclusion criteria, consider including a flowchart or a concise summary of the participant selection process. This will make it easier for readers to understand how the final sample was derived.
  2. Task Description: In the "The n-back task" section, provide more specific details about the n-back task, such as the number of trials, the duration of each trial, and the number of blocks in each condition (0-back and 2-back).
  3. Image Acquisition: Provide more specific details about the MRI acquisition parameters, such as field strength, sequence type, and any unique settings or considerations for each scanner type (Siemens, GE, Philips).
  4. Data Preprocessing: Explain the rationale for the preprocessing steps in more detail. For example, why were the first eight volumes excluded for some scanners, and what is the significance of a mean frame-wise displacement threshold of 0.3 mm?
  5. Statistical Analyses: Provide more clarity on the statistical analyses performed, especially in terms of the correction for multiple comparisons. Explain how the cluster-based method for multiple comparisons correction was applied and why specific thresholds were chosen.
  6. Figures and Tables: Consider including figures or tables that summarize the steps in your methodology. Visual aids can help readers grasp complex processes more effectively.
  7. Ethical Considerations: Mention ethical considerations, such as institutional review board (IRB) approval and informed consent procedures, to demonstrate the ethical rigor of your study.
  8. Variable Definitions: Define any abbreviations or acronyms used in the methods section to ensure clarity. For example, explain what "LI" stands for and how it is calculated.

Results

  1. Interpretation: While you present the results of your analyses, it would be helpful to provide some interpretation or context for these findings. Explain why certain results are important and how they contribute to your study's objectives.
  2. Subsection Headings: You can use subsection headings to further structure this section. For example, you can have subsections like "Demographic and Task Performance Results," "ROI-Based Activation Differences," and "Brain-Behavior Correlations" to provide a clear structure.
  3. Statistical Significance: When presenting p-values or statistical significance, it's a good practice to indicate the level of significance (e.g., p < 0.05) to help readers assess the importance of the findings.
  4. Consistency: Ensure consistency in the presentation of results. For example, if you use abbreviations (e.g., ADHD-F) in the text, make sure they are defined or explained the first time they appear.

Discussion  

  1. Interpretation of Findings: You have provided a comprehensive interpretation of your results, linking the differences in brain activation patterns to the symptoms and risk factors of ADHD. This is crucial for understanding the clinical significance of your findings.
  2. Comparison with Existing Literature: You've done a good job comparing your findings to existing literature, highlighting both consistencies and discrepancies. This helps contextualize your study within the broader research landscape.
  3. Sex Differences: You mentioned the inclusion of both male and female subjects and briefly discussed the potential for sex differences. Expanding on this point, even if your study didn't find significant sex-related differences, you can discuss the importance of considering sex as a variable in future research on ADHD.
  4. Network Analysis: Your suggestion to explore network analysis and topological features in future research is valuable. You can briefly outline how such an approach might provide a more comprehensive understanding of the neural basis of ADHD, especially in the context of familial risk.
  5. Longitudinal Research: You rightly mentioned the cross-sectional nature of your study and the need for longitudinal research. You can elaborate on the benefits of tracking the developmental trajectory of ADHD-related functional differences and how it might inform early intervention and treatment strategies.
  6. Limitations: You briefly touched on limitations, such as the age range of your participants. You could expand on this section by discussing other potential limitations, such as the sample size, recruitment methods, or the choice of neuropsychological tests and measures used.
  7. Clinical Implications: Conclude the discussion by summarizing the clinical implications of your findings. How might your results influence the diagnosis and treatment of ADHD, especially in children with a familial risk?
  8. Future Directions: End the discussion by highlighting specific directions for future research. What are the key unanswered questions that your study raises? How might researchers build upon your work to gain a deeper understanding of ADHD?

I wish to express my genuine gratitude for your submission of the article. I have meticulously examined its contents and have offered constructive feedback with the intention of elevating its overall quality. I kindly request that you consider implementing the proposed revisions and adjustments. Upon your completion of these revisions, I would be pleased to conduct a further review of the revised version!

Abstract

  1. Data and Sample Size: Mentioning the sample size (362 subjects) is crucial for transparency and context. However, it would be beneficial to include information about the age range of the participants and any other relevant demographic details.
  2. Hypotheses or Research Questions: The abstract should briefly outline the hypotheses or research questions that guided the study. This helps readers understand the purpose and focus of the research.
  3. Methods Overview: Provide a brief overview of the methods used in the study, such as the specific tasks used to assess working memory and the neuroimaging techniques employed (e.g., fMRI).
  4. Results Summary: While the abstract mentions the key findings, it could be more specific. For instance, it could briefly describe the direction and significance of the observed associations in the ADHD-F group (e.g., "significant positive association" with task reaction time) to give readers a better understanding of the results.
  5. Avoid Jargon: While some level of technical language is expected in an academic abstract, it's essential to strike a balance. Avoid overly technical jargon that may alienate readers who are not experts in the field.

Introduction

  1. Research Gap: Explicitly state the research gap or the problem that your study addresses. Why is it important to investigate familial vs. non-familial ADHD? What specific questions or hypotheses will your study answer? This will help readers understand the significance of your research.
  1. Demographic Details: While you mention that the ABCD Study replicates demographic characteristics of the general population, consider providing a concise summary of these characteristics, such as age, gender distribution, and any other relevant factors, to give readers a better understanding of the study population..
  2. Transition Sentences: Use transition sentences or phrases to smoothly connect different sections of the introduction. For example, after discussing the heritability of ADHD (paragraph starting with "Heritability estimates..."), you can transition into the next paragraph by explaining how familial risk relates to this heritability.
  3. Concept Definitions: Define key terms and concepts, such as "executive function," "working memory," and "neural correlates," to ensure that readers from various backgrounds can follow your argument without ambiguity.
  4. Rationale for the Study: Provide a more explicit rationale for why investigating working memory-related mechanisms in familial vs. non-familial ADHD is essential. How might these findings contribute to our understanding of ADHD etiology and treatment?

For enrichment of your intorductionpleas euse the below refences:
https://www.brainstimjrnl.com/article/S1935-861X(23)00171-7/fulltext

https://brieflands.com/articles/ijpbs-108390.html

Materials and Methods

  1. Participant Selection: While you provide information about participant exclusion criteria, consider including a flowchart or a concise summary of the participant selection process. This will make it easier for readers to understand how the final sample was derived.
  2. Task Description: In the "The n-back task" section, provide more specific details about the n-back task, such as the number of trials, the duration of each trial, and the number of blocks in each condition (0-back and 2-back).
  3. Image Acquisition: Provide more specific details about the MRI acquisition parameters, such as field strength, sequence type, and any unique settings or considerations for each scanner type (Siemens, GE, Philips).
  4. Data Preprocessing: Explain the rationale for the preprocessing steps in more detail. For example, why were the first eight volumes excluded for some scanners, and what is the significance of a mean frame-wise displacement threshold of 0.3 mm?
  5. Statistical Analyses: Provide more clarity on the statistical analyses performed, especially in terms of the correction for multiple comparisons. Explain how the cluster-based method for multiple comparisons correction was applied and why specific thresholds were chosen.
  6. Figures and Tables: Consider including figures or tables that summarize the steps in your methodology. Visual aids can help readers grasp complex processes more effectively.
  7. Ethical Considerations: Mention ethical considerations, such as institutional review board (IRB) approval and informed consent procedures, to demonstrate the ethical rigor of your study.
  8. Variable Definitions: Define any abbreviations or acronyms used in the methods section to ensure clarity. For example, explain what "LI" stands for and how it is calculated.

Results

  1. Interpretation: While you present the results of your analyses, it would be helpful to provide some interpretation or context for these findings. Explain why certain results are important and how they contribute to your study's objectives.
  2. Subsection Headings: You can use subsection headings to further structure this section. For example, you can have subsections like "Demographic and Task Performance Results," "ROI-Based Activation Differences," and "Brain-Behavior Correlations" to provide a clear structure.
  3. Statistical Significance: When presenting p-values or statistical significance, it's a good practice to indicate the level of significance (e.g., p < 0.05) to help readers assess the importance of the findings.
  4. Consistency: Ensure consistency in the presentation of results. For example, if you use abbreviations (e.g., ADHD-F) in the text, make sure they are defined or explained the first time they appear.

Discussion  

  1. Interpretation of Findings: You have provided a comprehensive interpretation of your results, linking the differences in brain activation patterns to the symptoms and risk factors of ADHD. This is crucial for understanding the clinical significance of your findings.
  2. Comparison with Existing Literature: You've done a good job comparing your findings to existing literature, highlighting both consistencies and discrepancies. This helps contextualize your study within the broader research landscape.
  3. Sex Differences: You mentioned the inclusion of both male and female subjects and briefly discussed the potential for sex differences. Expanding on this point, even if your study didn't find significant sex-related differences, you can discuss the importance of considering sex as a variable in future research on ADHD.
  4. Network Analysis: Your suggestion to explore network analysis and topological features in future research is valuable. You can briefly outline how such an approach might provide a more comprehensive understanding of the neural basis of ADHD, especially in the context of familial risk.
  5. Longitudinal Research: You rightly mentioned the cross-sectional nature of your study and the need for longitudinal research. You can elaborate on the benefits of tracking the developmental trajectory of ADHD-related functional differences and how it might inform early intervention and treatment strategies.
  6. Limitations: You briefly touched on limitations, such as the age range of your participants. You could expand on this section by discussing other potential limitations, such as the sample size, recruitment methods, or the choice of neuropsychological tests and measures used.
  7. Clinical Implications: Conclude the discussion by summarizing the clinical implications of your findings. How might your results influence the diagnosis and treatment of ADHD, especially in children with a familial risk?
  8. Future Directions: End the discussion by highlighting specific directions for future research. What are the key unanswered questions that your study raises? How might researchers build upon your work to gain a deeper understanding of ADHD?

Author Response

Manuscript ID: brainsci-2647904

Title: Working Memory-related Neurofunctional Correlates associated with Frontal Lobe in Children with Familial vs. Non-Familial Attention Deficit/Hyperactivity Disorder

We appreciate the very detailed and constructive comments from the reviewers. We have addressed all the comments and have made specific changes accordingly in the revised manuscript. Our point-by-point response to each comment is as follows in Red:

Reviewer 2

R2.1 – 2.5. Abstract

Data and Sample Size: Mentioning the sample size (362 subjects) is crucial for transparency and context. However, it would be beneficial to include information about the age range of the participants and any other relevant demographic details.

Hypotheses or Research Questions: The abstract should briefly outline the hypotheses or research questions that guided the study. This helps readers understand the purpose and focus of the research.

Methods Overview: Provide a brief overview of the methods used in the study, such as the specific tasks used to assess working memory and the neuroimaging techniques employed (e.g., fMRI).

Results Summary: While the abstract mentions the key findings, it could be more specific. For instance, it could briefly describe the direction and significance of the observed associations in the ADHD-F group (e.g., "significant positive association" with task reaction time) to give readers a better understanding of the results.

Avoid Jargon: While some level of technical language is expected in an academic abstract, it's essential to strike a balance. Avoid overly technical jargon that may alienate readers who are not experts in the field.

We appreciate these very detailed instructions in forming an Abstract with thorough information about the study. Unfortunately according to the formative requirements of Brain Sciences, we are only allowed to have a very brief and structured Abstract with no more than 200 words. Please see this requirement here (https://www.mdpi.com/journal/brainsci/instructions): “Abstract: The abstract should be a total of about 200 words maximum. The abstract should be a single paragraph and should follow the style of structured abstracts, but without headings: 1) Background: Place the question addressed in a broad context and highlight the purpose of the study; 2) Methods: Describe briefly the main methods or treatments applied. Include any relevant preregistration numbers, and species and strains of any animals used; 3) Results: Summarize the article's main findings; and 4) Conclusion: Indicate the main conclusions or interpretations. The abstract should be an objective representation of the article: it must not contain results which are not presented and substantiated in the main text and should not exaggerate the main conclusions.”

Due to the restriction of Journal Instructions, we are unable to fully address these comments within the text of Abstract. However, we tried our best to address as much as we could and have accommodated the rest in the main text in each section.

R2.6. (Introduction) Research Gap: Explicitly state the research gap or the problem that your study addresses. Why is it important to investigate familial vs. non-familial ADHD? What specific questions or hypotheses will your study answer? This will help readers understand the significance of your research.

We are sorry if we did not make these points clear enough in the original manuscript. The question why is it important to investigate familial vs. non-familial ADHD was addressed on page 2: “…Familial risk is an important etiological factor of ADHD. Heritability estimates are as high as 60%-80%, and studies have showed a 5-fold risk of developing ADHD and 4-fold risk of symptoms persistence into adolescence and/or adulthood in children with positive family history of the disorder [12-16]. Understanding the distinct neural mechanisms underlying familial, as compared to non-familial, ADHD is urgent and critical for development of more tailored diagnosis and neurobiologically targeted treatment/interventions in affected children…” The question about what specific questions or hypotheses will your study answer was addressed in the second paragraph of Page 3: “The current study aimed to investigate working memory-related functional brain mechanisms associated with familial vs. nonfamilial ADHD in three matched and independent groups of ADHD-F, ADHD-NF, and TDC from the baseline pool of the Adolescent Brain Cognitive Development (ABCD) Study. The ABCD Study recruitment replicated demographic characteristics of general population with similar rates of familial vs non-familial ADHD as shown by other large-scale community-based studies [13,14,45,46]. The baseline data of ABCD Study includes 9-10-years-old participants with 47.8% females and racial distribution including 52.1% White, 20.3% Hispanic, 15.0% Black, 2.1% Asian and others [47]. We hypothesized that relative to the groups of TDC and ADHD-NF, children with ADHD-F would show distinct activation patterns associated with the PFC during working memory process, and such neural alterations may be linked to cognitive impairment in working memory and/or ADHD symptomatology specific to ADHD-F. Therefore, the success of this proposed study may contribute to the development of distinct neural markers of ADHD-F vs. ADHD-NF that can inform more tailored diagnoses and treatments in affected children.” 

R2.7. (Introduction) Demographic Details: While you mention that the ABCD Study replicates demographic characteristics of the general population, consider providing a concise summary of these characteristics, such as age, gender distribution, and any other relevant factors, to give readers a better understanding of the study population.

Following the suggestion, we have added demographic information about the age range, sex and racial distribution which are relevant to the current study. Please see page 3: “…The baseline data of ABCD Study includes 9-10-years-old participants with 47.8% females and racial distribution including 52.1% White, 20.3% Hispanic, 15.0% Black, 2.1% Asian and others [47]

  1. Karcher, N.R.; Barch, D.M. The ABCD study: understanding the development of risk for mental and physical health outcomes. Neuropsychopharmacology 2021, 46, 131-142, doi:10.1038/s41386-020-0736-6.

R2.8. (Introduction) Transition Sentences: Use transition sentences or phrases to smoothly connect different sections of the introduction. For example, after discussing the heritability of ADHD (paragraph starting with "Heritability estimates..."), you can transition into the next paragraph by explaining how familial risk relates to this heritability.

We appreciate this valuable suggestion in soothing the paragraphs of a research article. However, the focus of this study is on the neural substrates associated with working memory deficits in ADHD-F vs. ADHD-NF. Further explorations in how familial risk relates to the heritability of ADHD is not closely relevant to the study scope. Therefore we take the next paragraphs, step-by-step, to review the existing studies that are more relevant to our study.

R2.9. (Introduction) Concept Definitions: Define key terms and concepts, such as "executive function," "working memory," and "neural correlates," to ensure that readers from various backgrounds can follow your argument without ambiguity.

Following this suggestion, we have defined these concepts on page 1: “…neural correlates (corresponding neural mechanisms of given cognitive function) …” and page 2: “EFs are a group of top-down mental processes responsible for focus and attention to the task at hand to aid in planning, inhibitory control and working memory [19]. Working memory is the interim storage and manipulation of necessary information for the execution of tasks at hand [20]

  1. Diamond, A. Executive functions. Annu Rev Psychol 2013, 64, 135-168, doi:10.1146/annurev-psych-113011-143750.
  2. Baddeley, A. Working memory. Science 1992, 255, 556-559, doi:10.1126/science.1736359.

R2.10. (Introduction) Rationale for the Study: Provide a more explicit rationale for why investigating working memory-related mechanisms in familial vs. non-familial ADHD is essential. How might these findings contribute to our understanding of ADHD etiology and treatment?

Following the suggestion, the complete second paragraph on page 2 now focuses on addressing the rationales of the study: Familial risk has been suggested to be an important contributor to impaired EF in ADHD children [33-35]. This suggestion is supported by twin and sibling studies in which reduced working memory performance (measured using verbal and visuo-spatial tasks) was observed in ADHD children/adolescents and their unaffected twin/siblings compared to TDC [14, 36-39]. These studies suggest an association between familial risk for ADHD and working memory deficits. Thus, investigating the neural substrates of working memory deficits associated with family risk of ADHD  may shed light on differential neural mechanisms underlying familial vs. non-familial ADHD (ADHD-F vs. ADHD-NF), which can further provide guidance on targeted pharmacological and non-pharmacological treatments in children with ADHD [40,41].

  1. Barkley, R.A. Behavioral inhibition, sustained attention, and executive functions: constructing a unifying theory of ADHD. Psychol Bull 1997, 121, 65-94, doi:10.1037/0033-2909.121.1.65.
  2. Brauer, H.; Ziegler, C.; Dempfle, A.; Freitag, C.; Siniatchkin, M.; Krauel, K.; Prehn-Kristensen, A. Transcranial direct current stimulation in ADHD–First results of the trial E-StimADHD. Brain Stimulation: Basic, Translational, and Clinical Research in Neuromodulation 2023, 16, 170-171.
  3. Khaksarian, M.; Mirr, I.; Kordian, S.; Nooripour, R.; Ahangari, N.; Masjedi-Arani, A. A comparison of methylphenidate (MPH) and combined methylphenidate with Crocus sativus (Saffron) in the treatment of children and adolescents with ADHD: A randomized, double-blind, parallel-group, clinical trial. Iranian Journal of Psychiatry and Behavioral Sciences 2021.

R2.11 (Introduction) For enrichment of your introduction please use the below refences: https://www.brainstimjrnl.com/article/S1935-861X(23)00171-7/fulltext, https://brieflands.com/articles/ijpbs-108390.html

We greatly appreciate this suggestion and have included these two references in the third paragraph of page 2, as references 40 and 41.

  1. Brauer, H.; Ziegler, C.; Dempfle, A.; Freitag, C.; Siniatchkin, M.; Krauel, K.; Prehn-Kristensen, A. Transcranial direct current stimulation in ADHD–First results of the trial E-StimADHD. Brain Stimulation: Basic, Translational, and Clinical Research in Neuromodulation 2023, 16, 170-171.
  2. Khaksarian, M.; Mirr, I.; Kordian, S.; Nooripour, R.; Ahangari, N.; Masjedi-Arani, A. A comparison of methylphenidate (MPH) and combined methylphenidate with Crocus sativus (Saffron) in the treatment of children and adolescents with ADHD: A randomized, double-blind, parallel-group, clinical trial. Iranian Journal of Psychiatry and Behavioral Sciences 2021.

R2.12. (Materials and Methods) Participant Selection: While you provide information about participant exclusion criteria, consider including a flowchart or a concise summary of the participant selection process. This will make it easier for readers to understand how the final sample was derived.

Following the suggestion, a summary paragraph is now provided on page 4: “Briefly, the participants of this study were obtained from the ABCD Study baseline pool. General inclusion/exclusion criteria were first applied to all the baseline subjects with or without ADHD. Imaging and task performance data quality were examined to further exclude unqualified subjects. Then among candidates in the group of ADHD, medical history of biological parents were assessed to form the sub-groups of ADHD-F and ADHD-NF and remove subjects with unclear family risk information. Participants in the TDC group were chosen to best match with the group of ADHD for age, sex, race, handedness, IQ, pubertal developmental scale, parental education, and combined income. Further information on participant characteristics can be found in Table 1.”

R2.13. (Materials and Methods) Task Description: In the "The n-back task" section, provide more specific details about the n-back task, such as the number of trials, the duration of each trial, and the number of blocks in each condition (0-back and 2-back).

We appreciate this comment and have supplied the missing details of the task on page 6.

R2.14. (Materials and Methods) Image Acquisition: Provide more specific details about the MRI acquisition parameters, such as field strength, sequence type, and any unique settings or considerations for each scanner type (Siemens, GE, Philips).

Following the suggestion, we have added the missing details on page 6, as per ABCD Study scan protocol.

R2.15. (Materials and Methods) Data Preprocessing: Explain the rationale for the preprocessing steps in more detail. For example, why were the first eight volumes excluded for some scanners, and what is the significance of a mean frame-wise displacement threshold of 0.3 mm?

Following the suggestion, we have included the additional explanations on page 7: “…because these initial volumes made up the pre-scan reference for the 4D images [62] mean frame-wise displacement > 0.3 mm which is an accepted and stringent cut-off for analyses based on ABCD Study cohort [60].”

  1. Casey, B.J.; Cannonier, T.; Conley, M.I.; Cohen, A.O.; Barch, D.M.; Heitzeg, M.M.; Soules, M.E.; Teslovich, T.; Dellarco, D.V.; Garavan, H.; et al. The Adolescent Brain Cognitive Development (ABCD) study: Imaging acquisition across 21 sites. Dev Cogn Neurosci 2018, 32, 43-54, doi:10.1016/j.dcn.2018.03.001.
  2. Hagler, D.J., Jr.; Hatton, S.; Cornejo, M.D.; Makowski, C.; Fair, D.A.; Dick, A.S.; Sutherland, M.T.; Casey, B.J.; Barch, D.M.; Harms, M.P.; et al. Image processing and analysis methods for the Adolescent Brain Cognitive Development Study. Neuroimage 2019, 202, 116091, doi:10.1016/j.neuroimage.2019.116091.

R2.16. (Materials and Methods) Statistical Analyses: Provide more clarity on the statistical analyses performed, especially in terms of the correction for multiple comparisons. Explain how the cluster-based method for multiple comparisons correction was applied and why specific thresholds were chosen.

We now added more details about cluster-based correction method and rationale for Z-threshold on page 7: “… This cluster-based method utilized Z-statistic threshold to define contiguous clusters, the significance level of which (from Gaussian random-field theory) was compared with probability threshold. The Z-threshold range from 2.3 to 3.1 corresponds to the primary threshold range of p-values from 0.01 to 0.001 and was therefore considered to be standard in fMRI studies [67].”

  1. Woo, C.W.; Krishnan, A.; Wager, T.D. Cluster-extent based thresholding in fMRI analyses: pitfalls and recommendations. Neuroimage 2014, 91, 412-419, doi:10.1016/j.neuroimage.2013.12.058.

R2.17. (Materials and Methods) Figures and Tables: Consider including figures or tables that summarize the steps in your methodology. Visual aids can help readers grasp complex processes more effectively.

Following the suggestion, Supplementary Figure 1 (in the Supplementary materials file) is now added to graphically display the steps in methodology, page 8: “…. Steps in methodology are summarized in Supplementary Figure 1.”

R2.18. (Materials and Methods) Ethical Considerations: Mention ethical considerations, such as institutional review board (IRB) approval and informed consent procedures, to demonstrate the ethical rigor of your study.

We appreciate this important comment. We have added the IRB statement and consent statement at the end of the manuscript.

R2.19. (Materials and Methods) Variable Definitions: Define any abbreviations or acronyms used in the methods section to ensure clarity. For example, explain what "LI" stands for and how it is calculated.

Thanks for this important suggestion. We have now defined LI, the equation for it, and its meaning.  We also have added the importance of laterality analyses in fMRI and ADHD prior to the equation of LI. Please see page 7: “Lateralized activation patterns for cognitive information processes have been frequently observed in fMRI studies in human subjects [73-76]. Laterality Index (LI) has been applied as a metric for describing the hemispherical domination patterns of fMRI activation [77,78]. In this study the LI...”

  1. Hull, R.; Vaid, J. Laterality and language experience. Laterality 2006, 11, 436-464, doi:10.1080/13576500600691162.
  2. Detre, J.A.; Maccotta, L.; King, D.; Alsop, D.C.; Glosser, G.; D'Esposito, M.; Zarahn, E.; Aguirre, G.K.; French, J.A. Functional MRI lateralization of memory in temporal lobe epilepsy. Neurology 1998, 50, 926-932, doi:10.1212/wnl.50.4.926.
  3. Hutsler, J.; Galuske, R.A. Hemispheric asymmetries in cerebral cortical networks. Trends Neurosci 2003, 26, 429-435, doi:10.1016/S0166-2236(03)00198-X.
  4. Strauss, E.; Kosaka, B.; Wada, J. The neurobiological basis of lateralized cerebral function. A review. Hum Neurobiol 1983, 2, 115-127.
  5. Seghier, M.L. Laterality index in functional MRI: methodological issues. Magn Reson Imaging 2008, 26, 594-601, doi:10.1016/j.mri.2007.10.010.
  6. He, N.; Palaniyappan, L.; Linli, Z.; Guo, S. Abnormal hemispheric asymmetry of both brain function and structure in attention deficit/hyperactivity disorder: a meta-analysis of individual participant data. Brain Imaging Behav 2022, 16, 54-68, doi:10.1007/s11682-021-00476-x.

R2.20. (Results) Interpretation: While you present the results of your analyses, it would be helpful to provide some interpretation or context for these findings. Explain why certain results are important and how they contribute to your study's objectives.

We appreciate this comment and understand the importance of contextual interpretation of the findings. Given the Journal Instructions for paper organizations that we need to follow, we have presented the interpretation, implications, and context of our findings in the discussion section (section 4).

R2.21. (Results) Subsection Headings: You can use subsection headings to further structure this section. For example, you can have subsections like "Demographic and Task Performance Results," "ROI-Based Activation Differences," and "Brain-Behavior Correlations" to provide a clear structure.

We appreciate this valuable suggestion and have included subsections of Results as follow:

3.1. Demographic and task-performance measures

3.2. ROI-based activation and LI analyses

3.3. Brain-behavior correlation analyses

R2.22. (Results) Statistical Significance: When presenting p-values or statistical significance, it's a good practice to indicate the level of significance (e.g., p < 0.05) to help readers assess the importance of the findings.

We have updated the manuscript by adding the p-values in the text. Please see from page 9: “…in left IFG (p<0.001, d=0.387) and bilateral SFG (left p=0.002, d=0.334; right p=0.006, d=0.29), whereas children with ADHD-NF showed significantly reduced mean activation in bilateral IFG (left p<0.001, d=0.367; right p<0.001, d=0.398) and right SFG (p=0.012, d=0.258…with the 2-back RT (p=0.003), the mean activation in the right IFG was significantly negatively correlated with CBCL ADHD T-score (p=0.021) and LI in IFG was significantly negatively correlated with KSADS-5 inattention raw score (p=0.001). In children with ADHD-NF, the LI of IFG showed significant positive correlation with the KSADS-5 hyperactivity raw score (p=0.037); the mean activation in left SFG showed significant negative correlation with CBCL attention problems T-score (p=0.025) …

R2.23. (Results) Consistency: Ensure consistency in the presentation of results. For example, if you use abbreviations (e.g., ADHD-F) in the text, make sure they are defined or explained the first time they appear.

We appreciate your valuable comment about consistency or presentation. We have checked for initial definitions for each acronym in the text and made sure that they are consistent.

R2.24. (Discussion) Interpretation of Findings: You have provided a comprehensive interpretation of your results, linking the differences in brain activation patterns to the symptoms and risk factors of ADHD. This is crucial for understanding the clinical significance of your findings.

We appreciate this very complimentary comment.

R2.25. (Discussion) Comparison with Existing Literature: You've done a good job comparing your findings to existing literature, highlighting both consistencies and discrepancies. This helps contextualize your study within the broader research landscape.

Again we appreciate this very complimentary comment.

R2.26. (Discussion) Sex Differences: You mentioned the inclusion of both male and female subjects and briefly discussed the potential for sex differences. Expanding on this point, even if your study didn't find significant sex-related differences, you can discuss the importance of considering sex as a variable in future research on ADHD.

Following the suggestion, we now further addressed this critical issue on page 12: “…Our study included both male and female subjects. Although it is unestablished yet whether the neurofunctional basis of familial ADHD have sex differences, differences in symptomatic and comorbidity profiles have previously been observed in clinical studies [104]. To partially omit the effects related to sex, we included sex as a fixed-factor covariate in group analyses after matching the groups for sex. Nevertheless, future research should explicitly assess the possibility that familial risk may affect boys and girls with ADHD differently.”

  1. Quinn, P.O.; Madhoo, M. A review of attention-deficit/hyperactivity disorder in women and girls: uncovering this hidden diagnosis. Prim Care Companion CNS Disord 2014, 16, doi:10.4088/PCC.13r01596.

R2.27. (Discussion) Network Analysis: Your suggestion to explore network analysis and topological features in future research is valuable. You can briefly outline how such an approach might provide a more comprehensive understanding of the neural basis of ADHD, especially in the context of familial risk.

We now added a statement explaining what kind of understanding can be obtained by network analysis approach in the context of familial ADHD. Please see page 12: “…The network-based approach can provide systems-level understanding of the inter-regional functional interactions during cognitive information processing and their association with ADHD-related deficits and familial risk…”

R2.28. (Discussion) Longitudinal Research: You rightly mentioned the cross-sectional nature of your study and the need for longitudinal research. You can elaborate on the benefits of tracking the developmental trajectory of ADHD-related functional differences and how it might inform early intervention and treatment strategies.

We now added a statement explaining how longitudinal study could be important to determine treatment targets of early onset in the trajectory of these deficits. Please see page 12: “…The outcomes of such studies can significantly inform the early brain markers for different developmental trajectories in children with ADHD, and guide early treatment and intervention strategies.

R2.29. (Discussion) Limitations: You briefly touched on limitations, such as the age range of your participants. You could expand on this section by discussing other potential limitations, such as the sample size, recruitment methods, or the choice of neuropsychological tests and measures used.

Following the suggestion, we have carefully amended the paragraph of limitations accordingly.

R2.30. (Discussion) Clinical Implications: Conclude the discussion by summarizing the clinical implications of your findings. How might your results influence the diagnosis and treatment of ADHD, especially in children with a familial risk?

The revised Section 5 (Conclusions) now fully addresses suggestions in this comment.

R2.31. (Discussion) Future Directions: End the discussion by highlighting specific directions for future research. What are the key unanswered questions that your study raises? How might researchers build upon your work to gain a deeper understanding of ADHD?

The lower half of the second paragraph of page 12 and the entire last paragraph of Discussions (pages 12-13) now fully address suggestions in this comment.

Reviewer 3 Report

Firstly, I am writing to express my gratitude for the opportunity to review the research article “Working Memory-related Neurofunctional Correlates associated with Frontal Lobe in Children with Familial vs. Non-Familial Attention Deficit/Hyperactivity Disorder”. I am honored to have been selected to contribute to the peer-review process for Brain Sciences.

I understand the critical importance of rigorous evaluation in academic research and am eager to lend my expertise to this process. I am confident that my analysis will be of value to the authors and help ensure that the work is of the highest quality.

Thank you for entrusting me with this important task. I look forward to the opportunity to provide a thorough and constructive review.

The study explores brain activation patterns related to working memory in children with ADHD, comparing those with and without a family history of ADHD to typically developing children. Both ADHD groups showed reduced activation in a specific brain region compared to typically developing children. The group with a family history of ADHD had unique associations between brain activation and task performance and ADHD symptoms. This suggests potential for tailored diagnosis and interventions for ADHD in children with a family history of the disorder.

I would like to make a series of improvement suggestions to the authors:

INTRODUCCIÓN

  • Enhanced Understanding of Heterogeneity: Elaborate on the sources of heterogeneity in ADHD, considering additional factors such as comorbidities or developmental stages.

  • Comprehensive Overview of ADHD Symptomatology: Provide a comprehensive overview of ADHD symptomatology, considering the domains of inattention, hyperactivity, and impulsivity.

  • Dive into Executive Functioning: Elaborate on various facets of executive functions beyond working memory, like inhibition, cognitive flexibility, and planning, as these are interconnected and important in ADHD.

  • Clarify the Inconsistent Evidence on EF Deficits: Discuss potential reasons for inconsistencies in evidence regarding executive function deficits in ADHD, including methodological differences across studies and the heterogeneity of the disorder.

  • Address Potential Methodological Biases: Address possible biases or limitations in previous studies that might have influenced the heterogeneity of results, such as variations in age, comorbidity rates, or sample sizes.

METHOD

Participant Recruitment and Selection:

Specify the rationale for choosing the ABCD Study and the advantages of its large participant pool for this investigation.

ADHD Symptomatology Assessment:

Provide more information on the validation and reliability of the KSADS-5 in assessing ADHD symptomatology.

ADHD Subtype Determination:

Clarify the methods and criteria used for determining ADHD presentation (Inattentive, Hyperactive-Impulsive, Combined) based on the Diagnostic and Statistical Manual (version 5) criteria.

Image Acquisition Specifics:

Specify the exact number of subjects for whom fMRI and sMRI data were successfully acquired, as this is vital for understanding the study sample.

Voxel-Based Activation Maps Thresholding:

Explain the choice of Z ≥ 2.3 threshold for voxel-based activation maps and its relevance in the context of fMRI analysis.

ROI Selection and Definition Rationale:

Clarify the rationale for selecting ROI pairs based on between-group differences and how these regions were deemed relevant to the study.

Laterality Analyses Explanation:

Provide a brief explanation of why the Laterality Index (LI) is important for studying activation patterns and how it's calculated.

Brain-Behavioral Correlation Rationale:

Explain the rationale behind correlating ROI activation and laterality with ADHD symptomatology and task-performance measures and how this aids in understanding the neurofunctional correlates.

RESULTS

Detailed Presentation of Significant Differences:

Clearly present the effect sizes or magnitude of differences in task-performance measures between ADHD-F and ADHD-NF groups to better understand the practical implications of the observed significant differences.

Expand on ROI Analysis:

Provide a more comprehensive analysis of the ROI-based activation patterns, potentially exploring sub-regions within the ROIs to identify any nuanced differences in activation.

Further Explanation of Lateralization Trends:

Elaborate on the potential implications and theories behind the trend towards right-hemisphere-shifted Laterality Index (LI) in the ADHD-F group, providing insights into the neural mechanisms associated with this lateralization pattern.

Correlation Analysis Interpretation:

Provide a detailed interpretation of the correlations observed in the brain-behavior analysis, elucidating the implications of these correlations on working memory and ADHD symptomatology in both ADHD-F and ADHD-NF groups.

Integration of Previous Research Findings:

Integrate the current findings with existing literature to provide context and discuss how the current study contributes to the understanding of working memory-related neurofunctional correlates in children with ADHD.

DISCUSSION

Address Sex Differences and Impact on Results:

Discuss the potential impact of sex differences on the neurofunctional basis of familial ADHD and highlight the importance of future research explicitly addressing this aspect to better understand any sex-related variations in the observed patterns.

Quantify and Discuss Effect Sizes:

Quantify and discuss effect sizes for significant findings to provide a clearer understanding of the magnitude of the observed effects, enhancing the interpretation of the results and their practical implications.

Expand on Limitations and Future Research Directions:

Elaborate on the limitations of the study in more detail, addressing potential confounding factors and their impact on the results. Additionally, suggest specific future research directions to address these limitations.

Discuss Clinical Relevance and Translational Potential:

Discuss the clinical relevance of the findings and their potential to serve as biomarkers for tailored diagnoses and targeted treatments, emphasizing how these findings could translate into improved interventions and outcomes for children with ADHD.

Provide a Comprehensive Summary:

Summarize the key findings and their implications for both familial and non-familial ADHD, ensuring a clear and concise conclusion that ties together the entire discussion and emphasizes the main takeaways from the research.

Continue the excellent effort; your commitment and diligence are clearly reflected in the caliber of your research. I am confident that, with some refinements, this manuscript will be prepared for submission.

Reviewing your work has been a delightful experience, and I am assured that, with the proposed revisions, your paper will significantly enrich the field. I extend my best wishes for your ongoing research endeavors and eagerly anticipate your forthcoming publications.

I'd like to convey my heartfelt gratitude for the dedication and hard work you've put into your research. The article stands to benefit from substantial enhancements, and I strongly recommend that the authors undertake a comprehensive revision to elevate its quality before submitting it again.

Regards,

Author Response

Manuscript ID: brainsci-2647904

Title: Working Memory-related Neurofunctional Correlates associated with Frontal Lobe in Children with Familial vs. Non-Familial Attention Deficit/Hyperactivity Disorder

We appreciate the very detailed and constructive comments from the reviewers. We have addressed all the comments and have made specific changes accordingly in the revised manuscript. Our point-by-point response to each comment is as follows in Red:

Reviewer 3

R3.1 - 3.5 (Introduction)

Enhanced Understanding of Heterogeneity: Elaborate on the sources of heterogeneity in ADHD, considering additional factors such as comorbidities or developmental stages.

Comprehensive Overview of ADHD Symptomatology: Provide a comprehensive overview of ADHD symptomatology, considering the domains of inattention, hyperactivity, and impulsivity.

Dive into Executive Functioning: Elaborate on various facets of executive functions beyond working memory, like inhibition, cognitive flexibility, and planning, as these are interconnected and important in ADHD.

Clarify the Inconsistent Evidence on EF Deficits: Discuss potential reasons for inconsistencies in evidence regarding executive function deficits in ADHD, including methodological differences across studies and the heterogeneity of the disorder.

Address Potential Methodological Biases: Address possible biases or limitations in previous studies that might have influenced the heterogeneity of results, such as variations in age, comorbidity rates, or sample sizes.

We appreciate these detailed instructions and have amended these parts of introduction in multiple places on pages 1- 2 according: “…Clinically, ADHD may present as primarily inattentive, primarily hyperactive/impulsive, or combined type when diagnosed based on at least 6-month presence of symptoms in multiple settings such as school and home [4,5].” “… ADHD children show a notable heterogeneity in clinical and cognitive/behavioral profiles, comorbidities, developmental stages and trajectories, and neural correlates (underlying neural mechanisms associated with the given cognitive function), likely due to heterogeneity in etiological factors (i.e., biological and environmental risk factors) [1,6-8].” “ …Considerable evidence on multiple units of analysis (e.g., behavior, paradigms, physiology, circuits) have separately implicated each of the executive function (EF) and attention domains in the pathophysiology of ADHD, while the moderate effect size of these findings and the substantial proportion of children with ADHD in these studies who did not exhibit impairments prove that individual deficits in these cognitive domains cannot account for all cases of the disorder [9-11]…” And “... EFs are a group of top-down mental processes responsible for focus and attention to the task at hand to aid in planning, inhibitory control and working memory [19]. Working memory is the interim storage and manipulation of necessary information for the execution of tasks at hand [20]…”

  1. Baeyens, D.; Roeyers, H.; Walle, J.V. Subtypes of attention-deficit/hyperactivity disorder (ADHD): distinct or related disorders across measurement levels? Child Psychiatry Hum Dev 2006, 36, 403-417, doi:10.1007/s10578-006-0011-z.
  2. Chhabildas, N.; Pennington, B.F.; Willcutt, E.G. A comparison of the neuropsychological profiles of the DSM-IV subtypes of ADHD. J Abnorm Child Psychol 2001, 29, 529-540, doi:10.1023/a:1012281226028.
  3. Willcutt, E.G.; Doyle, A.E.; Nigg, J.T.; Faraone, S.V.; Pennington, B.F. Validity of the executive function theory of attention-deficit/hyperactivity disorder: a meta-analytic review. Biol Psychiatry 2005, 57, 1336-1346, doi:10.1016/j.biopsych.2005.02.006.
  4. Nigg, J.T.; Willcutt, E.G.; Doyle, A.E.; Sonuga-Barke, E.J. Causal heterogeneity in attention-deficit/hyperactivity disorder: do we need neuropsychologically impaired subtypes? Biol Psychiatry 2005, 57, 1224-1230, doi:10.1016/j.biopsych.2004.08.025.
  5. Yap, K.H.; Abdul Manan, H.; Sharip, S. Heterogeneity in brain functional changes of cognitive processing in ADHD across age: A systematic review of task-based fMRI studies. Behav Brain Res 2021, 397, 112888, doi:10.1016/j.bbr.2020.112888.
  6. Diamond, A. Executive functions. Annu Rev Psychol 2013, 64, 135-168, doi:10.1146/annurev-psych-113011-143750.
  7. Baddeley, A. Working memory. Science 1992, 255, 556-559, doi:10.1126/science.1736359.

R3.6. (Methods) Participant Recruitment and Selection: Specify the rationale for choosing the ABCD Study and the advantages of its large participant pool for this investigation.

We have added the rationale in the introduction section and the advantages of large dataset in the methods section. Section 1, page 3: “…The baseline data of ABCD Study includes 9-10-years-old participants with 47.8% females and racial distribution including 52.1% White, 20.3% Hispanic, 15.0% Black, 2.1% Asian and others [47]…  Section 2.1, page 3: “…This large sample size provides increased statistical power to compare groups and draw conclusions…”

  1. Karcher, N.R.; Barch, D.M. The ABCD study: understanding the development of risk for mental and physical health outcomes. Neuropsychopharmacology 2021, 46, 131-142, doi:10.1038/s41386-020-0736-6.

R3.7. (Methods) ADHD Symptomatology Assessment: Provide more information on the validation and reliability of the KSADS-5 in assessing ADHD symptomatology.

We have added a statement emphasizing the validity and reliability of KSADS-5 in multiple settings. Section 2.1, page 3: “…KSADS-5 has been rigorously studied for validation and reliability in multiple research and clinical settings among children and adolescents [54,55]…”

  1. Kariuki, S.M.; Newton, C.; Abubakar, A.; Bitta, M.A.; Odhiambo, R.; Phillips Owen, J. Evaluation of Psychometric Properties and Factorial Structure of ADHD Module of K-SADS-PL in Children From Rural Kenya. J Atten Disord 2020, 24, 2064-2071, doi:10.1177/1087054717753064.
  2. de la Pena, F.R.; Villavicencio, L.R.; Palacio, J.D.; Felix, F.J.; Larraguibel, M.; Viola, L.; Ortiz, S.; Rosetti, M.; Abadi, A.; Montiel, C.; et al. Validity and reliability of the kiddie schedule for affective disorders and schizophrenia present and lifetime version DSM-5 (K-SADS-PL-5) Spanish version. BMC Psychiatry 2018, 18, 193, doi:10.1186/s12888-018-1773-0.

R3.8. (Methods) ADHD Subtype Determination: Clarify the methods and criteria used for determining ADHD presentation (Inattentive, Hyperactive-Impulsive, Combined) based on the Diagnostic and Statistical Manual (version 5) criteria.

We have added specific criteria for each presentation type. Section 2.1, page 4: “…Subjects who were endorsed for 6 or more symptoms out of 9 on inattention symptom count were considered as inattentive presentation, 6 or more symptoms out of 9 on hyperactivity symptom count considered as hyperactive/impulsive presentation and 6 or more symptoms out of 9 on both domains considered as combined presentation of ADHD...”

R3.9. (Methods) Image Acquisition Specifics: Specify the exact number of subjects for whom fMRI and sMRI data were successfully acquired, as this is vital for understanding the study sample.

We have added the total number of subjects in the image acquisition section. Section 2.3, page 4: “…fMRI data for each of the 362 subjects were acquired…”

R3.10. (Methods) Voxel-Based Activation Maps Thresholding: Explain the choice of Z ≥ 2.3 threshold for voxel-based activation maps and its relevance in the context of fMRI analysis.

We have added the rationale of Z-threshold range from 2.3 to 3.1 in fMRI studies. Section 2.4, page 7: “…This cluster-based method utilized Z-statistic threshold to define contiguous clusters, the significance level of which (from Gaussian random-field theory) was compared with probability threshold. The Z-threshold range from 2.3 to 3.1 corresponds to the primary threshold range of p-values from 0.01 to 0.001 and was therefore considered to be standard in fMRI studies [67].”

  1. Woo, C.W.; Krishnan, A.; Wager, T.D. Cluster-extent based thresholding in fMRI analyses: pitfalls and recommendations. Neuroimage 2014, 91, 412-419, doi:10.1016/j.neuroimage.2013.12.058.

R3.11. (Methods) ROI Selection and Definition Rationale: Clarify the rationale for selecting ROI pairs based on between-group differences and how these regions were deemed relevant to the study.

We have added the rationale for PFC regions and voxel-based group differences in the methods section. Section 2.5, page 7: “The PFC is one of the core components of the brain pathways for working memory processing, and has been frequently reported to involve in the neurophysiology of ADHD [68]. Voxel-based fMRI studies have reported functional deficits in multiple clusters of PFC in children with ADHD [69]. Based on results of these existing studies and the voxel-based results of the current study, three bilateral ROI-pairs were identified in regions of PFC....”

  1. Martinussen, R.; Hayden, J.; Hogg-Johnson, S.; Tannock, R. A meta-analysis of working memory impairments in children with attention-deficit/hyperactivity disorder. J Am Acad Child Adolesc Psychiatry 2005, 44, 377-384, doi:10.1097/01.chi.0000153228.72591.73.
  2. Davis, T.; LaRocque, K.F.; Mumford, J.A.; Norman, K.A.; Wagner, A.D.; Poldrack, R.A. What do differences between multi-voxel and univariate analysis mean? How subject-, voxel-, and trial-level variance impact fMRI analysis. Neuroimage 2014, 97, 271-283, doi:10.1016/j.neuroimage.2014.04.037.

R3.12. (Methods) Laterality Analyses Explanation: Provide a brief explanation of why the Laterality Index (LI) is important for studying activation patterns and how it's calculated.

We have added the importance of laterality analyses in fMRI and ADHD prior to the equation of LI. Section 2.6, page 7-8: “Lateralized activation patterns for cognitive information processes have been frequently observed in fMRI studies in human subjects [73-76]. Laterality Index (LI) has been applied as a metric for describing the hemispherical domination patterns of fMRI activation [77,78]...”

  1. Hull, R.; Vaid, J. Laterality and language experience. Laterality 2006, 11, 436-464, doi:10.1080/13576500600691162.
  2. Detre, J.A.; Maccotta, L.; King, D.; Alsop, D.C.; Glosser, G.; D'Esposito, M.; Zarahn, E.; Aguirre, G.K.; French, J.A. Functional MRI lateralization of memory in temporal lobe epilepsy. Neurology 1998, 50, 926-932, doi:10.1212/wnl.50.4.926.
  3. Hutsler, J.; Galuske, R.A. Hemispheric asymmetries in cerebral cortical networks. Trends Neurosci 2003, 26, 429-435, doi:10.1016/S0166-2236(03)00198-X.
  4. Strauss, E.; Kosaka, B.; Wada, J. The neurobiological basis of lateralized cerebral function. A review. Hum Neurobiol 1983, 2, 115-127.
  5. Seghier, M.L. Laterality index in functional MRI: methodological issues. Magn Reson Imaging 2008, 26, 594-601, doi:10.1016/j.mri.2007.10.010.
  6. He, N.; Palaniyappan, L.; Linli, Z.; Guo, S. Abnormal hemispheric asymmetry of both brain function and structure in attention deficit/hyperactivity disorder: a meta-analysis of individual participant data. Brain Imaging Behav 2022, 16, 54-68, doi:10.1007/s11682-021-00476-x.

R3.13. (Methods) Brain-Behavioral Correlation Rationale: Explain the rationale behind correlating ROI activation and laterality with ADHD symptomatology and task-performance measures and how this aids in understanding the neurofunctional correlates.

We have briefly included the importance of associating functional activations to these measures. Section 2.7, page 8: “Understanding the brain-behavioral relationship is critical for searching the neural mechanisms associated with cognitive impairments and clinical symptoms in brain disorders [79]…”

  1. Buracas, G.T.; Fine, I.; Boynton, G.M. The relationship between task performance and functional magnetic resonance imaging response. J Neurosci 2005, 25, 3023-3031, doi:10.1523/JNEUROSCI.4476-04.2005.

R3.14. (Results) Detailed Presentation of Significant Differences: Clearly present the effect sizes or magnitude of differences in task-performance measures between ADHD-F and ADHD-NF groups to better understand the practical implications of the observed significant differences.

We have added the effect sizes along with each significant result in the results section for task-performance and ROI activation differences. For correlation analysis, Pearson’s correlation coefficient for each result is displayed in Figure 4.

R3.15. (Results) Expand on ROI Analysis: Provide a more comprehensive analysis of the ROI-based activation patterns, potentially exploring sub-regions within the ROIs to identify any nuanced differences in activation.

We appreciate this comment and would like to apologize if we did not make it clear in our original manuscript. The choice of these ROIs was intentionally made in accordance with the rationale outlined in the main text (page 7), by utilizing the standard masks from the AAL atlas version 3. According to the established fundamental principles in neuroscience, these AAL-based ROIs in the prefrontal cortex do not contain functionally meaningful subregions any longer. Therefore, explorations in sub-regions of these ROIs would lack support from neuroscience foundations and it would be hard to interpret any results of such analyses.

R3.16 – 18. (Results)

Further Explanation of Lateralization Trends: Elaborate on the potential implications and theories behind the trend towards right-hemisphere-shifted Laterality Index (LI) in the ADHD-F group, providing insights into the neural mechanisms associated with this lateralization pattern.

Correlation Analysis Interpretation: Provide a detailed interpretation of the correlations observed in the brain-behavior analysis, elucidating the implications of these correlations on working memory and ADHD symptomatology in both ADHD-F and ADHD-NF groups.

Integration of Previous Research Findings: Integrate the current findings with existing literature to provide context and discuss how the current study contributes to the understanding of working memory-related neurofunctional correlates in children with ADHD.

We are grateful for these suggestions. However, according to the Author’s Instructions, the section of Results is supposed to provide results of analyses listed in the section of Methodology. The suggestions of further explaining the lateralization trends, correlation analysis interpretation, and integration of previous research findings, have been substantially addressed in the section of Discussions.

R3.19. (Discussion) Address Sex Differences and Impact on Results: Discuss the potential impact of sex differences on the neurofunctional basis of familial ADHD and highlight the importance of future research explicitly addressing this aspect to better understand any sex-related variations in the observed patterns.

We appreciate this comment and have addressed it in Section 4, page 12: “…Our study included both male and female subjects. Although it is unestablished yet whether the neurofunctional basis of familial ADHD have sex differences, differences in symptomatic and comorbidity profiles have previously been observed in clinical studies [104]. To partially omit the effects related to sex, we included sex as a fixed-factor covariate in group analyses after matching the groups for sex. Nevertheless, future research should explicitly assess the possibility that familial risk may affect boys and girls with ADHD differently. …”

  1. Quinn, P.O.; Madhoo, M. A review of attention-deficit/hyperactivity disorder in women and girls: uncovering this hidden diagnosis. Prim Care Companion CNS Disord 2014, 16, doi:10.4088/PCC.13r01596.

R3.20. (Discussion) Quantify and Discuss Effect Sizes: Quantify and discuss effect sizes for significant findings to provide a clearer understanding of the magnitude of the observed effects, enhancing the interpretation of the results and their practical implications.

We appreciate this important and valuable suggestion. We have added details on effect sizes in the Results and Discussion sections.

R3.21. (Discussion) Expand on Limitations and Future Research Directions: Elaborate on the limitations of the study in more detail, addressing potential confounding factors and their impact on the results. Additionally, suggest specific future research directions to address these limitations.

Discussions on the limitations of the current study and future directions are now substantially extended in the revised manuscript. Please see page 12-13 for details.

R3.22 - 23. (Discussion)

Discuss Clinical Relevance and Translational Potential: Discuss the clinical relevance of the findings and their potential to serve as biomarkers for tailored diagnoses and targeted treatments, emphasizing how these findings could translate into improved interventions and outcomes for children with ADHD.

Provide a Comprehensive Summary: Summarize the key findings and their implications for both familial and non-familial ADHD, ensuring a clear and concise conclusion that ties together the entire discussion and emphasizes the main takeaways from the research.

We appreciate this comment and believe that the entire section of Conclusions has been focused on summarizing the key findings and their implications, discussing the clinical relevance of the findings and their potential to serve as biomarkers for tailored diagnoses and targeted treatments, emphasizing how these findings could translate into improved interventions and outcomes for children with ADHD, and concluding the main takeaways of the research. 

Round 2

Reviewer 2 Report

The revisions to your paper have been diligently implemented, resulting in a fully corrected and now acceptable manuscript. We extend our gratitude for your dedicated efforts and collaborative approach in elevating the overall quality of your article.

We wish you continued success in your forthcoming research and writing endeavors.

The revisions to your paper have been diligently implemented, resulting in a fully corrected and now acceptable manuscript. We extend our gratitude for your dedicated efforts and collaborative approach in elevating the overall quality of your article.

We wish you continued success in your forthcoming research and writing endeavors.